



# 1 Liquid-liquid phase separation and viscosity within secondary organic

# 2 aerosol generated from diesel fuel vapors

Mijung Song[1,2], Adrian M. Maclean[2], Yuanzhou Huang[2], Natalie R. Smith[3], Sandra L. Blair[3],
Julia Laskin[4], Alexander Laskin[4], Wing-Sy Wong DeRieux[3], Ying Li[3], Manabu Shiraiwa[3],
Sergey A. Nizkorodov[3], Allan K. Bertram[2*]
[1] {Department of Earth and Environmental Sciences, Chonbuk National University,
Jeollabuk-do, 54896, Republic of Korea}
[2] {Department of Chemistry, University of British Columbia, Vancouver, BC, V6T 1Z1,
Canada}
[3] {Department of Chemistry, University of California Irvine, Irvine, CA 92697, USA}
[4] {Department of Chemistry, Purdue University, Wes Lafayette, IN 47907, USA}
**Abstract**
Information on liquid-liquid phase separation (LLPS) and viscosity (or diffusion) within
secondary organic aerosol (SOA) is needed to improve predictions of particle size, mass,
reactivity, and cloud nucleating properties in the atmosphere. Here we report on LLPS and
viscosities within SOA generated by the photooxidation of diesel fuel vapors. Diesel fuel
contains a wide range of volatile organic compounds, and SOA generated by the photooxidation
of diesel fuel vapors may be a good proxy for SOA from anthropogenic emissions. In our
experiments, LLPS occurred over the relative humidity (RH) range of ~70 % to ~100 %,
resulting in an organic-rich outer phase and a water-rich inner phase. These results may have
implications for predicting the cloud nucleating properties of anthropogenic SOA since the
organic-rich outer phase can lower the kinetic barrier for activation to a cloud droplet. At ≤ 10 %
RH, the viscosity was ≥ $1{\times}10^8$ Pa s, which corresponds to roughly the viscosity of tar pitch. At
38 - 50 % RH the viscosity was in the range of $1{\times}10^8$ - $3{\times}10^5$ Pa s. These measured viscosities
are consistent with predictions based on oxygen to carbon elemental ratio (O:C) and molar
mass as well as predictions based on the number of carbon, hydrogen, and oxygen atoms. Based
on the measured viscosities and the Stokes-Einstein relation, at ≤ 10 % RH diffusion
coefficients of organics within diesel fuel SOA is ≤ $5.4{\times}10^{-17}$ cm$^2$ s$^{-1}$ and the mixing time of
organics within 200 nm diesel fuel SOA particles ($\tau_{mixing}$) is ≳ 50 h. These small diffusion



coefficients and large mixing times may be important in laboratory experiments, where SOA
is often generated and studied using low RH conditions and on time scales of minutes to hours.
At 38 - 50 % RH, the calculated organic diffusion coefficients are in the range of $5.4 \times 10^{-17}$ to
$1.8 \times 10^{-13}$ cm$^2$ s$^{-1}$ and calculated $\tau_{mixing}$ values are in the range of ~0.01 h to ~50 h. These values
provide important constraints for the physicochemical properties of anthropogenic SOA.
**1 Introduction**
Volatile organic compounds (VOCs) are emitted into the atmosphere from both biogenic and
anthropogenic sources (Kanakidou et al., 2005; Hallquist et al., 2009). These VOCs can be
oxidized in the atmosphere, and the oxidized products can form secondary organic aerosol
(SOA) (Hallquist et al., 2009; Ervens et al., 2011). SOA accounts for 20 – 80 % of the mass of
atmospheric aerosol particles (Zhang et al., 2007; Jimenez et al., 2009) and plays an important
role in climate, air quality, and public health (Kanakidou et al., 2005; Jang et al., 2006; Solomon,
2007; Baltensperger et al., 2008; Murray et al., 2010; Wang et al., 2012; Pöschl and Shiraiwa,
2015; Shiraiwa et al., 2017; Shrivastava et al., 2017). Despite the importance of SOA, many of
the physicochemical properties of SOA remain poorly understood.
One physicochemical property of SOA that remains insufficiently understood is liquid-liquid
phase separation (LLPS) (Pankow, 2003; Marcolli and Krieger, 2006; Ciobanu et al., 2009;
Bertram et al., 2011; Krieger et al., 2012; Song et al., 2012a; Zuend and Seinfeld, 2012; Veghte
et al., 2013; You et al., 2014; O'Brien et al., 2015; Freedman, 2017). Very recent work has
shown that SOA particles free of inorganic salts can undergo LLPS at a high relative humidity
(RH) with implications for predicting the cloud nucleating properties of SOA (Petters et al.,
2006; Hodas et al., 2016; Renbaum-Wolff et al., 2016; Ovadnevaite et al., 2017; Rastak et al.,
2017; Song et al., 2017; Altaft et al., 2018; Liu et al., 2018; Song et al., 2018; Davies et al.,
2019). Several of these recent studies investigated SOA generated from a single VOC (e.g. α-
pinene or isoprene). However, in the atmosphere, SOA is formed from a complex mixture of
VOCs (Odum et al., 1997; Schauer et al., 2002a; 2002b; Vutukuru et al., 2006; Velasco et al.,
2007; de Gouw et al., 2008; Velasco et al., 2009; Gentner et al., 2012; Liu et al., 2012; Hayes
et al., 2015). Additional studies are needed to determine if SOA generated from a complex
mixture of VOCs of atmospheric relevance can also undergo LLPS at high RH.
Another physicochemical property of SOA that remains poorly understood is viscosity.
Viscosity together with the Stokes-Einstein equation can be used to predict diffusion rates of

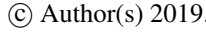



organics within SOA, which can critically impact a number of processes involving SOA. For
example, diffusion of organics within SOA can impact particle size distributions (Shiraiwa et
al., 2013a; Zaveri et al., 2014; Zaveri et al., 2018) and particle mass concentrations (Shiraiwa
and Seinfeld, 2012; Ye et al., 2016; Yli-Juuti et al., 2017; Kim et al., 2019) in the atmosphere.
Diffusion rates within SOA can also affect multi-phase reactions (Shiraiwa et al., 2011; Zhou
et al., 2013; Steimer et al., 2014; Houle et al., 2015; Li et al., 2018), the extent of long-range
transport of pollutants (Zelenyuk et al., 2012; Zhou et al., 2013; Shrivastava et al., 2017; Mu
et al., 2018), ice nucleation (Murray et al., 2010; Wang et al., 2012; Wilson et al., 2012; Ladino
et al., 2014; Schill et al., 2014; Knopf et al., 2018), and crystalline of salts (Murray, 2008;
Murray and Bertram, 2008; Bodsworth et al., 2010; Song et al., 2013; Ji et al., 2017; Wang et
al., 2017).
Recently, a number of studies have investigated viscosity or diffusion rates within SOA
particles generated in the laboratory (Virtanen et al., 2010; Cappa et al., 2011; Perraud et al.,
2012; Saukko et al., 2012; Abramson et al., 2013; Robinson et al., 2013; Renbaum-Wolff et al.,
2013; Loza et al., 2013; Kidd et al., 2014; Pajunoja et al., 2014; Bateman et al., 2015; Li et al.,
2015; Song et al., 2015; Wang et al., 2015; Zhang et al., 2015; Grayson et al., 2016; Liu et al.,
2016; Song et al., 2016a; Ullmann et al., 2019; Ye et al., 2018). Almost all of these studies
focused on SOA generated from a single VOC. Additional studies that quantify the viscosity
of SOA generated from a complex mixture of VOCs of atmospheric relevance are also needed.
Functional group contribution methods have recently been used to predict viscosities within
organic matrices of atmospheric relevance (Song et al., 2016a; Song et al., 2016b; Grayson et
al., 2017; Rothfuss and Petters, 2017). Methods have also been developed to predict the glass
transition temperature and viscosity within an organic matrix of atmospheric relevance using
molar mass and oxygen to carbon elemental ratio (O:C) (Shiraiwa et al., 2017) or the number
of carbon, hydrogen and oxygen atoms of the organic compounds within the organic matrix
(DeRieux et al., 2018). These methods, if accurate, should be useful for predicting viscosity of
SOA particles in the atmosphere.
Diesel fuel contains a wide range of VOCs including aromatics and alkanes. Furthermore, SOA
generated from the photooxidation of diesel fuel vapors may be a good proxy for SOA from
anthropogenic emissions (Odum et al., 1997; Schauer et al., 2002a; 2002b; Vutukuru et al.,
2006; Velasco et al., 2007; Velasco et al., 2009; de Gouw et al., 2008; Gentner et al., 2012; Liu
et al., 2012; Jathar et al., 2013; Jathar et al., 2014; Hayes et al., 2015; Blair et al., 2017; Gentner





1. et al., 2017; Jathar et al., 2017). In this study, we investigate LLPS and viscosity within SOA

2. particles generated by photooxidation of diesel fuel vapors. Measured viscosities are also

3. compared with predicted viscosities based on the methods developed by Shiraiwa et al. (2017)

4. and DeRieux et al. (2018). Based on the measured viscosities and the Stokes-Einstein relation,

5. diffusion coefficients and mixing times of large organic molecules within diesel fuel SOA is

6. also estimated.

7.

8. **2  Experimental**

9. **2.1 SOA generation**

10. SOA from the photooxidation of diesel fuel vapors was produced in an identical manner to that

11. described previously (DSL/NO$_x$ in Table 1 of Blair et al. (2017)). 45 uL of H$_2$O$_2$ (30 wt %) was

12. evaporated in a 5.6 m$^3$ inflatable Teflon chamber to achieve a mixing ratio of 2 parts per million

13. by volume (ppmv). A mixture of NO in N$_2$ was injected from a gas cylinder to achieve 0.26

14. ppmv of NO in the chamber. A volume of 200 uL of Fluka. No. 2 diesel (UST-148, 50 mg mL$^-$

15. $^1$ solution of diesel in dichloromethane) was evaporated in the chamber, resulting in a

16. concentration of 1.8 mg m$^{-3}$ organic vapor from diesel and a mixing ratio of 0.22 ppmv (based

17. on an average molecular weight of 200 g mol$^{-1}$ (Blair et al., 2017) and assuming no wall loss).

18. No seed aerosol was used, and the chamber RH was below 2%. UV-B lamps (FS40T12/UVB,

19. Solarc Systems Inc.) were used to drive the photooxidation, which lasted for 3 h, followed by

20. particle collection. After 3 h of photooxidation, the particle mass loading in the chamber was

21. 550 µg m$^{-3}$ based on measurements with a scanning mobility particle sizer (SMPS; TSI 3080

22. Electrostatic Classifier and TSI 3775 Condensation Particle Counter). An Aerodyne time-of-

23. flight aerosol mass spectrometer (ToF-AMS) was used to measure the particle mass spectra in

24. V mode. ToF-AMS data was analyzed using Squirrel version 1.61. For elemental analysis we

25. relied on the improved-ambient method by Canagaratna et al. (2015). Figure S1 shows typical

26. particle number concentration, mass concentration, and average atomic ratios during the

27. photooxidation. The O:C values (0.4 to 0.5) were consistent with O:C values reported by Blair

28. et al. (2017) for identically prepared samples.

29. For the LLPS and viscosity measurements, the SOA from the chamber was collected on

30. hydrophobic glass slides (12 mm coverslips, Hampton Research, Canada) for 120 min using

31. an inertial impactor for 120 min. To make the surface of the glass slides hydrophobic, they

32. were coated with trichloro(1*H*,1*H*,2*H*,2*H*-perfluorooctyl)silane (Sigma-Aldrich) following the



procedure reported in Knopf (2003). After collection, the sizes of the SOA particles on the
hydrophobic glass slides were > 10 μm. These large sizes were formed by impaction and
coagulation of the SOA during collection.

**2.2. Measurements of LLPS**

LLPS within the collected SOA particles was determined using a flow-cell with temperature
and RH control coupled to an optical microscope (Zeiss Epiplan 10X/0.20 HD) (Parsons et al.,
2004; Pant et al., 2006; Song et al., 2012b). A constant flow (1.5 L min$^{-1}$) of humidified $N_2$ gas
was maintained within the flow cell throughout the experiments. The RH of the humidified $N_2$
gas, measured with a dew point hygrometer (General Eastern M4/E4 Dew Point Monitor,
Canada), was varied from 100% to close to 0% during the experiments. The temperature within
the flow-cell measured with a thermocouple (OMEGA, Canada) was maintained at 290 ± 1 K.
For the LLPS experiments, first, the SOA particles were equilibrated at around 100% RH for
at least 15 min. Next, the RH was reduced a rate of 0.5% RH min$^{-1}$ until a value close to 0%
was reached. While the RH was decreased, images of the particles were acquired every 10 sec
with a CCD camera connected to the microscope. From the images, the number of phases (e.g.
one liquid phase or two liquid phases) present in the particles were determined.

**2.3 Measurements of particle viscosity**

The viscosity of the collected particles was determined using the poke-and-flow technique,
which has been described by Renbaum-Wolff et al. (2013) and Grayson et al. (2015), and based,
in part, on the earlier experiments by Murray et al. (2012). In short, the SOA particles collected
on hydrophobic glass slides were placed inside a flow-cell with RH and temperature control
(Pant et al., 2006; Bertram et al., 2011; Song et al., 2012a). After conditioning the particles to
a known RH at 294 ± 1 K, the particles were poked with a sharp needle (~10 μm for the tip of
the needle) (Becton-Dickson, USA). The movement of the needle was controlled with a
micromanipulator (Narishige, model MO-202U, Japan). The change in morphology as a
function of time after poking the particles with the needle was recorded with a camera attached
to the microscope. From the morphology changes and fluid dynamics simulations, upper and
lower limits to the SOA viscosity were determined. Fluid dynamics simulations were
performed using the finite-element analysis software package, *COMSOL Multiphysics*



(Renbaum-Wolff et al., 2013; Grayson et al., 2015). The geometry used in the simulations was
based on the geometry of the particles after poking them with a needle. Additional details of
the poke-and-flow experiments and the fluid dynamics simulations are discussed in Sect. 3.2
and Sect. S1-S3 of the Supplement.
**2.4 Predictions of viscosity based on high-resolution mass spectrometry**
Viscosities of the diesel fuel SOA was predicted using the elemental composition of the SOA
and the methods developed by Shiraiwa et al. (2017) and DeRieux et al. (2018). The elemental
compositions of the diesel fuel SOA were taken from a previous study (Blair et al., 2017) using
of SOA generated with identical conditions (DSL/NO$_x$ Table 1 of Blair et al. (2017)). In the
previous study by Blair et al. (2017) high-resolution nanospray desorption electrospray
ionization mass spectrometry (Roach et al., 2010) was used to determine the elemental
composition.
Shiraiwa et al. (2017) reported a parameterization (Eq. 1) to estimate the glass transition
temperature ($T_g$) of individual CH or CHO compounds with molar mass < ~450 g mol$^{-1}$.
$T_g = A + BM + CM^2 + D\,(O{:}C) + E\,M\,(O{:}C)$        (1)
where $M$ is the molar mass and O:C is the ratio of oxygen to carbon atoms. The coefficients
are: A = -21.57 (K), B = 1.51 (K mol g$^{-1}$), C = -1.7×10$^{-3}$ (K mol$^2$ g$^{-2}$), D = 131.4 (K) and E = -
0.25 (K mol g$^{-1}$).
DeRieux et al. (2018) reported another parameterization (Eq. 2) to predict $T_g$ of CH and CHO
compounds with molar mass up to ~1100 g mol$^{-1}$ using the number of carbon ($n_C$), hydrogen
($n_H$), and oxygen atoms ($n_O$):
$T_g = (n_C^0 + ln(n_C))\,b_C + ln(n_H)\,b_H + ln(n_C)\,ln(n_H)\,b_{CH} + ln(n_O)\,b_O + ln(n_C)\,ln(n_O)\,b_{CO}$        (2)
Values of the coefficients [$n_C^0$, $b_C$, $b_H$, $b_{CH}$, $b_O$, and $b_{CO}$] are [1.96, 61.99, -113.33, 28.74, 0, 0]
for CH compounds and [12.13, 10.95, -41.82, 21.61, 118.96, -24.38] for CHO compounds
(DeRieux et al., 2018).
To estimate the $T_g$ for a dry organic mixture ($T_{g,org}$), the relative mass concentration of each
compound was assumed to be proportional to its relative abundance in the mass spectrum and




the Gordon-Taylor mixing rule was employed with a Gordon-Taylor coefficient ($k_{GT}$) value of
1 (Dette et al., 2014).
For the $T_g$ of a mixture of organics and water ($T_{g,mix}$), the effective hygroscopicity parameter
($\kappa$) was applied to calculate the mass fraction of water in the SOA particles (Petters and
Kreidenweis, 2007). A $\kappa$ value of 0.1 was used for the diesel fuel SOA based on an average
O:C of 0.45 for diesel fuel-derived SOA (Fig.S1 and Table S2 in Blair et al. (2017)) and the
relationship between O:C and $\kappa$ reported in Lambe et al. (2011, Fig. 7) and Massoli et al. (2010,
Fig. 2). To estimate the $T_{g,mix}$, the Gordon-Taylor equation was applied with $k_{GT}$ set to 2.5,
based on previous studies that suggested 2.5 ± 1.0 (Zobrist et al., 2008; Koop et al., 2011;
Berkemeier et al., 2014).
Once $T_{g,mix}$ was determined, viscosity was estimated using the modified Vogel-Tammann-
Fulcher (VTF) equation and an assumed viscosity of $10^{12}$ Pa s at the glass transition
temperature ($T = T_g$) and an assumed viscosity of $10^{-5}$ Pa s at a very high temperature (Angell,
1991; Angell, 2002):
$$\log \eta = -5 + 0.434 \frac{T_0 D_f}{T - T_0} \qquad (3)$$
where   $$T_0 = \frac{39.17\, T_g}{D_f + 39.17} \qquad (4)$$
In these equations, $D_f$ is the fragility parameter and $T_0$ is the Vogel temperature. In our
calculations, we assume $D$ equal to 10 based on previous studies that showed a lower limit of
~10 (±1.7) as the molar mass increases (DeRieux et al., 2018).
**3  Results and discussion**
**3.1 LLPS in diesel fuel SOA**
Figures 1 and S2 show examples of images recorded during the LLPS experiments as the RH
was decreased from ~100% to ~0%. At the highest RH values (~100%), two phases were
observed in all cases. The inner phase was most likely a water-rich phase while the outer phase
was likely an organic-rich phase since the inner phase decreased in size as the RH decreased.
This conclusion is consistent with surface tensions of organics and experiments that have
investigated morphology of particles after LLPS (Jasper, 1972; Kwamena et al., 2010; Reid et



al., 2011; Song et al., 2013; O'Brien et al., 2015; Gorkowski et al., 2016, 2017). The organic-
rich phase was most likely non-crystalline since SOA contains thousands of molecules and the
concentration of any individual molecule is likely below the concentration required for
crystallization (Marcolli et al., 2004). At ~70 % RH, two liquid phases remained in most
particles (Figs. 1 and S2). In the few cases where LLPS was not clearly observed at ~70 % RH
(Figs. S2a and S2b), two liquid phases may still have been present in the particles, but not in
the focus of the microscope. Small amounts of the water-rich phase were present even at ≲ 50 %
RH in some cases (Figs. 1b and S2c).
In the previous studies using SOA derived from a single VOC, LLPS was observed when the
average O:C was between 0.34 and 0.44 but not when the average O:C was between 0.52 and
1.30 (Renbaum-Wolff et al., 2016, Rastak et al., 2017, Song et al., 2017). Consistent with this
trend, in the current studies, we observed LLPS when the O:C values of the SOA was 0.4 - 0.5
(Fig. S1b). However, in the previous studies using SOA derived from a single VOC, LLPS was
only observed between ~95 % and close to ~100 % RH. Whereas, in the current study, LLPS
was observed between ~70 % and close to ~100 %, in most cases. This suggests that as the
complexity of SOA increases, LLPS can occur over a wider range of RH values. Consistent
with this conclusion, in a recent study, we showed that LLPS in organic particles containing
two commercially available organic compounds occurs over a wider RH range than in particles
containing only one organic compound (Song et al., 2018). The increase in the range of RH
values over which LLPS occurs is likely related to the spread in O:C values within the organic
particles − as the spread in O:C values increases, the RH range for LLPS is also likely to
increase. Additional studies focusing on the spread of O:C values in SOA and the connection
with LLPS would be useful.
**3.2 Viscosity of diesel fuel-derived SOA**
**3.2.1 Lower limits to viscosity at 10% RH**
In these experiments, the RH was first decreased to 10% and particles were conditioned at this
RH for approximately 1 h. After conditioning, the particles were poked with a needle, which
caused the particles to crack (Fig. 2a). After poking, the sharp edges that resulted from cracking
moved by less than 0.5 μm in 5 h. The distance of 0.5 μm corresponds to the minimum amount
of movement that could be discerned in our microscope setup. Based on these results and fluid





dynamics simulations (Sect. S1 in the Supplement), the lower limit to the viscosity at 10 % RH
is $1\times10^8$ Pa s (Fig. 3a). This corresponds to roughly the viscosity of tar pitch (Koop et al., 2011).
**3.2.2. Lower limits to viscosity at 31 and 50 % RH**
In these experiments, the RH was first decreased to 31 % or 50 %, and conditioned at these RH
values for 1 h and 0.5 h, respectively. After conditioning the particles at either 31 or 50 % RH,
they were poked with a needle, resulting in the formation of a half torus geometry (Figs. 2b
and 2c). From images recorded after poking the particles, the experimental flow time, $\tau_{exp,\ flow}$,
was determined, which corresponds to the time for the equivalent-area diameter of the inside
of the half torus geometry to reduce by 50 %. The equivalent-area diameter, $d$, was calculated
via the relationship $d = (4A/\pi)^{1/2}$ where $A$ is the hole area (Reist, 1992). Based on the measured
$\tau_{exp,\ flow}$ values and fluid dynamics simulations (Renbaum-Wolff et al., 2013; Grayson et al.,
2015), and Sect. S2 in the Supplement, the lower limit to the viscosity is approximately $3\times10^4$
and $8\times10^5$ Pa s at 50 % and 31 % RH, respectively (Fig. 3a). For reference, the viscosity of
peanut butter corresponds is approximately $10^3$ Pa s (Koop et al., 2011).
**3.2.3. Upper limits to viscosity at RH values ranging from 38 to 60 %**
In these experiments, the following new procedure was used. First, the particles were exposed
to a dry nitrogen flow at 0 % RH for ~1 h. After this exposure, the particles were poked with a
needle resulting in cracking of the particles. The RH above the particles was then increased in
a single step to one of the following RH values: 38 %, 41 %, 48 %, 53 %, 57 %, and 60 %. As
the RH increased and then stabilized (which took 5-10 min), the cracked particles began to
flow and returned to an approximately spherical cap shape (e.g. Fig. 4). From images recorded
during these experiments, the time required for the particles to return to a spherical cap shape
(starting from the cracked particles at RH= 0%) was determined. This time (which included
the time for the RH to increase and stabilize) was referred to as the experimental recovery time,
$\tau_{exp,recovery}$. Based on the $\tau_{exp,recovery}$ values and fluid dynamics simulations (Sect. S3 in the
Supplement), the upper limits of the viscosity is $\sim1\times10^7$ Pa s and $\sim1\times10^8$ Pa s at RH values of
60 % and 38 %, respectively (Fig. 3a).
**3.2.4 Comparison with previous measurements and predictions**
In Fig. 3b the measured viscosities determined from individual poke-and-flow experiments are
grouped by RH and compared with the viscosity of SOA generated by the photooxidation of
toluene. Toluene SOA is commonly used as a proxy of anthropogenic SOA (Pandis et al., 1992;





Robinson et al., 2013; Bateman et al., 2015; Liu et al., 2016; Song et al., 2016). The viscosities
of the toluene SOA and the diesel fuel SOA are similar. At RH values between 38 and 50 %
both have viscosities in the range of approximately $10^4$ to $10^8$ Pa s while at ≤ 10 % RH, both
have viscosities ≥ $1 \times 10^8$ Pa s.
In Fig. 3b, the viscosity of diesel fuel SOA is also compared with predicted viscosities based
on O:C and molar mass (Eq. 1) and the number of carbon, hydrogen, and oxygen atoms (Eq.
2). Within the uncertainty of the measurements, the predicted viscosities are consistent with
the measured viscosities (Fig. 3b). Measurements of viscosity with reduced uncertainties would
be useful to better test the predictions. Common methods used to measure viscosities (i.e., bulk
viscometers) are more precise than the poke-and-flow technique, but require more material
than is typically produced in environmental chambers (Reid et al., 2018).
Interestingly, predictions based on the number of carbon, hydrogen, and oxygen atoms (Eq. 2)
are almost 3 orders of magnitude higher than predictions based on O:C and molar mass (Eq. 1)
for dry conditions (i.e. 0 % RH) (Fig. 3b). Eq. 2 was applied to molar masses up to ~1100 g
$mol^{-1}$ while Eq. 1 was applied to molar masses < 450 g $mol^{-1}$. If Eq. 2 was limited to molar
mass < 450 g $mol^{-1}$, the predicted viscosities would only decrease by a factor of ≤ 1.3 (Fig. S5).
The difference in the predictions based on Eq. 2 and Eq. 1 shown in Fig. 3b is due to the
uncertainties in those two parameterizations. More comprehensive experimental $T_g$ datasets are
needed to further refine the $T_g$ parameterizations.
The predicted viscosities shown in Fig. 3b only consider CH and CHO compounds. For the
diesel fuel SOA studied here, 257 compounds (~36% of the intensity weighted peaks) were
CHON compounds (Blair et al., 2017). A comprehensive experimental $T_g$ dataset for organic
compounds containing nitrogen atoms is required to improve the viscosity predictions of diesel
fuel SOA.
**3.3 Diffusion coefficients and mixing times of large organics within diesel fuel SOA**
From the measured viscosities, we calculated diffusion coefficients of the organic molecules
within the diesel fuel SOA using the Stokes-Einstein relation:
$$D_{org} = \frac{kT}{6\pi a \eta} \qquad\qquad (5)$$



Where $k$ is the Boltzmann constant, $T$ is the temperature, $a$ is the hydrodynamic radius of the
diffusing species, and $\eta$ is the dynamic viscosity. To calculate diffusion coefficients, we
assumed a hydrodynamic radius of 0.4 nm for the diffusing organic molecules (Renbaum-Wolff
et al., 2013). Although the Stokes-Einstein relation may under predict diffusion of small
molecules (e.g., OH, $O_3$, $NO_x$, $NH_3$, and $H_2O$) in SOA, this equation gives reasonable values
when the size of the diffusing organics is similar to the size of the matrix molecules and the
temperature is not too close to the $T_g$ of the matrix (Champion et al., 2000; Marshall et al., 2016;
Price et al., 2015, 2016; Bastelberger et al., 2017; Chenyakin et al., 2017; Ullmann et al., 2019).
Based on the measured viscosities and the Stokes-Einstein relation, the diffusion coefficients
of organics within Diesel SOA is $\leq 5.4 \times 10^{-17}$ cm$^2$ s$^{-1}$ for RH values $\leq 10\%$ (Fig. 5a, secondary
$y$-axis). For RH values between 38 % and 50 %, the diffusion coefficients are in the range of
$5.4 \times 10^{-17}$ to $1.8 \times 10^{-13}$ cm$^2$ s$^{-1}$.
From the calculated $D_{org}$, the mixing time of organics within 200 nm diesel fuel SOA particles,
$\tau_{mixing}$, was calculated with the following equation (Seinfeld and Pandis, 2006; Shiraiwa et al.,

15   2011):

$$\tau_{mixing} = \frac{d^2}{4\pi D_{org}} \qquad\qquad (6)$$
Where $d$ corresponds to the diameter of the SOA particles. Values of $\tau_{mixing}$ represent the time
after which the concentration of the diffusing molecules at the center of the particles deviates
by less than $e^{-1}$ from the equilibrium concentration. When calculating $\tau_{mixing}$, we assumed $d$ was
200 nm, which is consistent with the median diameter of the volume distribution of SOA in the
atmosphere (Martin et al., 2010; Pöschl et al., 2010; Riipinen et al., 2011).
It is often assumed in chemical transport models that organic molecules are well mixed in SOA
on the time scale of 1 h. Based on our viscosity results and Eq. 6, $\tau_{mixing}$ is $\gtrsim 50$ h at $\leq 10$ %
RH (Fig. 5a, secondary $y$-axis). This mixing time is much larger than assumed in chemical
transport models. However, in the planetary boundary layer, the RH in not often $\leq 10$ %, at
least not when SOA concentrations are significant (Fig. 5b and 5c). Nevertheless, the large
$\tau_{mixing}$ values at $\leq 10$ % RH, may be important in laboratory experiments, where SOA is often





generated and studied under low RH conditions on the time scales of minutes to hours. At 30 %
RH $\tau_{mixing}$ is $\gtrsim$0.4 h, and at 38 to 50 % RH $\tau_{mixing}$ is in the range of ~0.01 h to ~50 h (Fig. 5a).
These results provide important constraints on $\tau_{mixing}$ values within anthropogenic SOA.
Several caveats apply to the calculated $\tau_{mixing}$ values. First, the diesel fuel SOA was generated
using relatively high particle mass concentrations (~500 μg m$^{-3}$). The viscosity of diesel fuel
SOA may be higher if generated using lower particle mass concentrations (Grayson et al., 2016;
Jain et al., 2018). Second, $\tau_{mixing}$-values may be overestimated at low RH values due to the
possible breakdown of the Stokes-Einstein relation near the glass transition RH (Champion et
al., 2000; Bastelberger et al., 2017; Chenyakin et al., 2017; Evoy et al., 2019; Ullmann et al.,

10     2019).

**4    Summary and conclusions**
We investigated LLPS in SOA generated from diesel fuel vapors. Diesel fuel contains a wide
range of VOCs, and diesel fuel SOA may be a reasonable proxy for SOA from anthropogenic
emissions. Two liquid phases (an organic-rich outer phase and a water-rich inner phase) were
observed in the diesel fuel SOA at RH values ranging from ~70 % to ~100 %. These results
may be important for predicting the cloud nucleating ability of anthropogenic SOA since the
presence of an organic-rich outer phase at high RH values can lower the barrier to cloud droplet
formation (Petters et al. 2006; Hodas et al. 2016; Renbaum-Wolff et al., 2016; Rastak et al.,
2017; Ovadnevaite et al., 2017; Liu et al., 2018). The presence of two liquid phases at RH
values as low as ~70 % may also impact heterogeneous chemistry, growth, and optical
properties of SOA (Zuend et al., 2010; Zuend and Seinfeld, 2012; Shiraiwa et al., 2013b;
Freedman, 2017; Fard et al., 2018; Zhang et al., 2018). We conclude that LLPS should be
considered when predicting the cloud nucleating ability, reactivity, growth, and optical
properties of SOA from anthropogenic emissions.
We also investigated the viscosity of diesel fuel SOA using the poke-and-flow technique
together with simulations of fluid flow. For RH values of $\leq$ 10 %, the viscosity was $\geq 1 \times 10^8$
Pa s. At RH values between 30 and 50 % the viscosity was in the range of $1 \times 10^8$ to $3 \times 10^4$ Pa
s. The measured viscosities were consistent with predictions based on molar mass and O:C and
predictions based on the number of carbon, hydrogen, and oxygen atoms of identified SOA
compounds. Additional measurements of viscosity of diesel fuel SOA with reduced





uncertainties would be useful to better test the predictions. Furthermore, additional
comprehensive experimental $T_g$ datasets are needed to further refine the parameterizations.
Based on these measured viscosities and the Stokes-Einstein relation, diffusion coefficients and
$\tau_{mixing}$ values of organics within diesel fuel SOA particles were calculated. For RH values ≤10%,
diffusion coefficients are $\leq 5.4 \times 10^{-17}$ cm$^2$ s$^{-1}$ and $\tau_{mixing}$ is $\gtrsim$ 50 h. Such low RH values are not
common in the planetary boundary layer, but are common in laboratory experiments when
generating SOA. We conclude that these large $\tau_{mixing}$ should be considered when interpreting
laboratory data of SOA generated under low RH conditions. For RH values between 38 % and
50 %, the diffusion coefficients are in the range of $5.4 \times 10^{-17}$ to $1.8 \times 10^{-13}$ cm$^2$ s$^{-1}$ and $\tau_{mixing}$
values are in the range of ~0.01 h and ~50 h. These results provide important constraints on
diffusion coefficients and $\tau_{mixing}$ values within anthropogenic SOA. Further studies are needed
using more atmospherically relevant mass concentrations since a relatively high mass
concentration (~500 µg m$^{-3}$) of the SOA was used when generating the SOA in this work.
**Conflicts of interest**
There are no conflicts of interest to declare.
**Author contributions**
A.K.B designed the study. M.Song, A.M.M, and Y. H. performed the viscosity and LLPS
experiments. S.A.N., N.R.S., S.L.B., J. L., and A.L. generated the SOA and analyzed their
chemical compositions. W.-S.W.D., Y.L., and M.S. predicted viscosities. M.Song and A.K.B.
prepared the manuscript with contributions from all co-authors.
**Acknowledgements**
This work was supported by the Natural Sciences and Engineering Research Council of Canada.
M. Song acknowledges funding from the National Research Foundation of Korea (NRF), the
Korea Government (MSIP) (2016R1C1B1009243) and Korea Institute of Toxicology (KIT)
(KK-1905). M.S. acknowledges funding from the U.S. National Science Foundation (AGS-
1654104) and the U.S. Department of Energy (DE-SC0018349). The AMS instrument used in
this work was acquired with the NSF grant MRI-0923323.

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





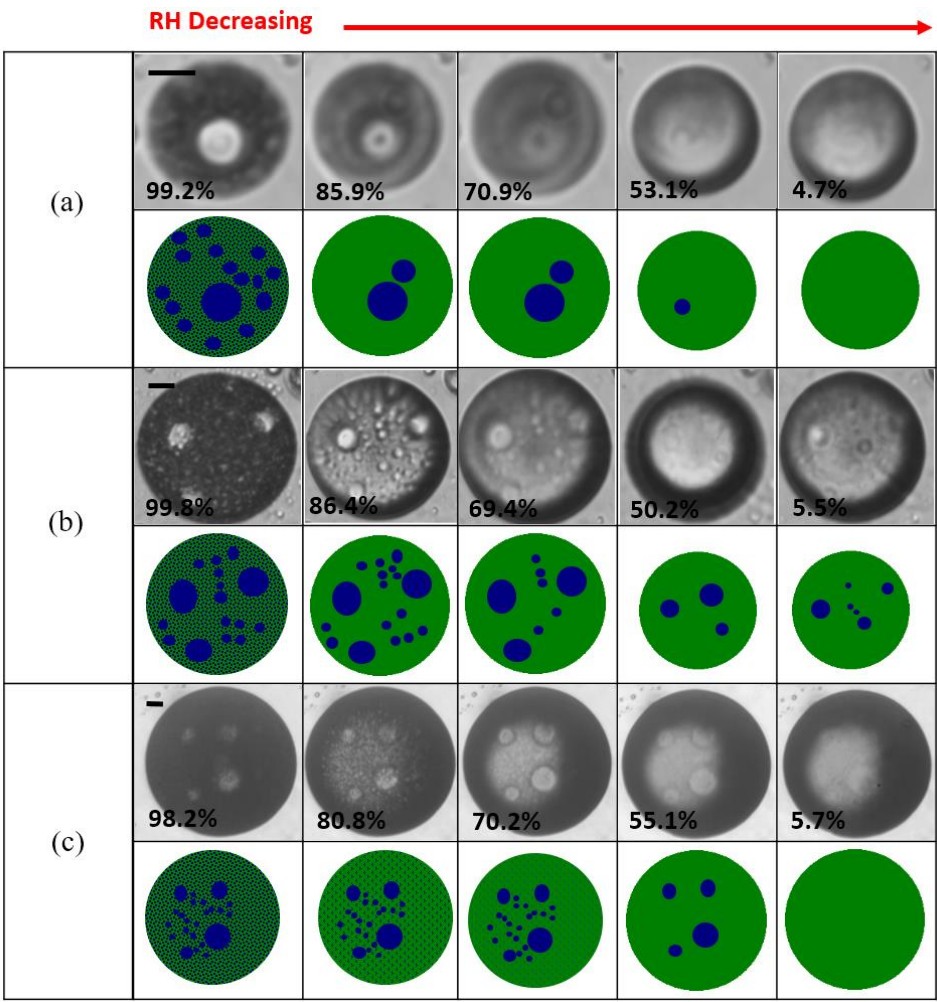

Figure 1. Optical images and illustrations of three diesel fuel SOA particles taken while the RH
was decreased. Illustrations are provided to help interpret the optical images with green color
representing the organic-rich phase, and blue color representing the water-rich phase. The
numbers under the optical images indicate the RH. The length of the scale bar is 10 μm.





Figure 2. Optical images of SOA particles during a poke-and-flow experiment: (a) 10 % RH,
(b) 31 % RH, and (c) 50 % RH. The size of the scale bar is 20 μm.

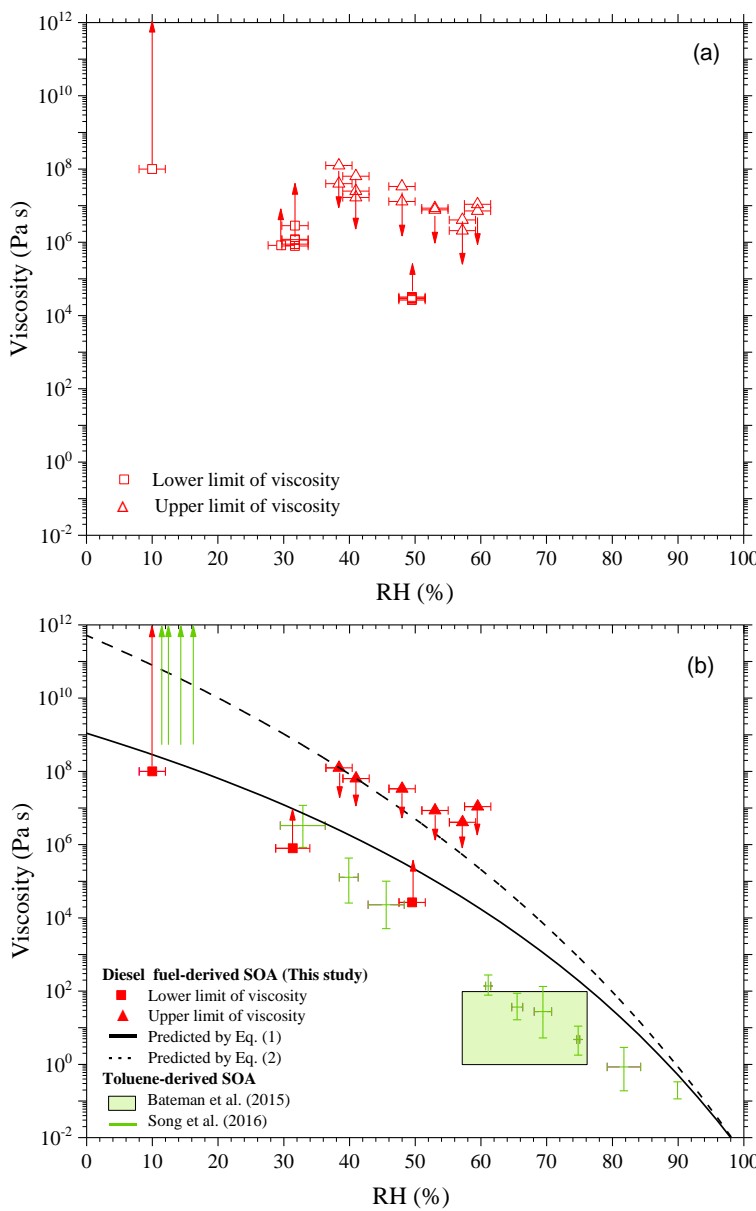

3 Figure 3. (a) Viscosities of diesel fuel SOA. Each data point corresponds to a viscosity

4 determined from a single poke-and-flow experiment. Upward arrows indicate lower limit to

5 the viscosities and downward arrows indicate upper limit to the viscosities of diesel fuel SOA.

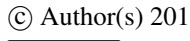



The $x$ error bars represent uncertainty in the RH measurements. (b) Viscosities of diesel fuel-
derived SOA but with the viscosities from individual poke-and-flow experiments grouped by
RH. The lower limit to the viscosities and the upper limit to the viscosities represent the lowest
and the highest viscosities in the group, respectively. At least two data points were included in
each group. The $x$ error bars represent the lowest and highest RH ranges in the group and the
uncertainty in the RH measurements. Also included are viscosities of toluene SOA from
Bateman et al. (2015) (green box) and Song et al. (2016) (green bar) and predicted viscosities
of the diesel fuel SOA using Eq. (1) (black solid line) and Eq. (2) (black dashed line).



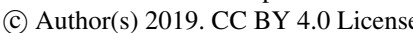



Figure 4. Optical images of diesel fuel SOA particles during poke-and-flow experiments.    In these experiments the SOA particles were poked at 0 % RH and then exposed to RH values of 53% (a) and 38% (b). The last column shows the particles after they have returned to a spherical cap shape.

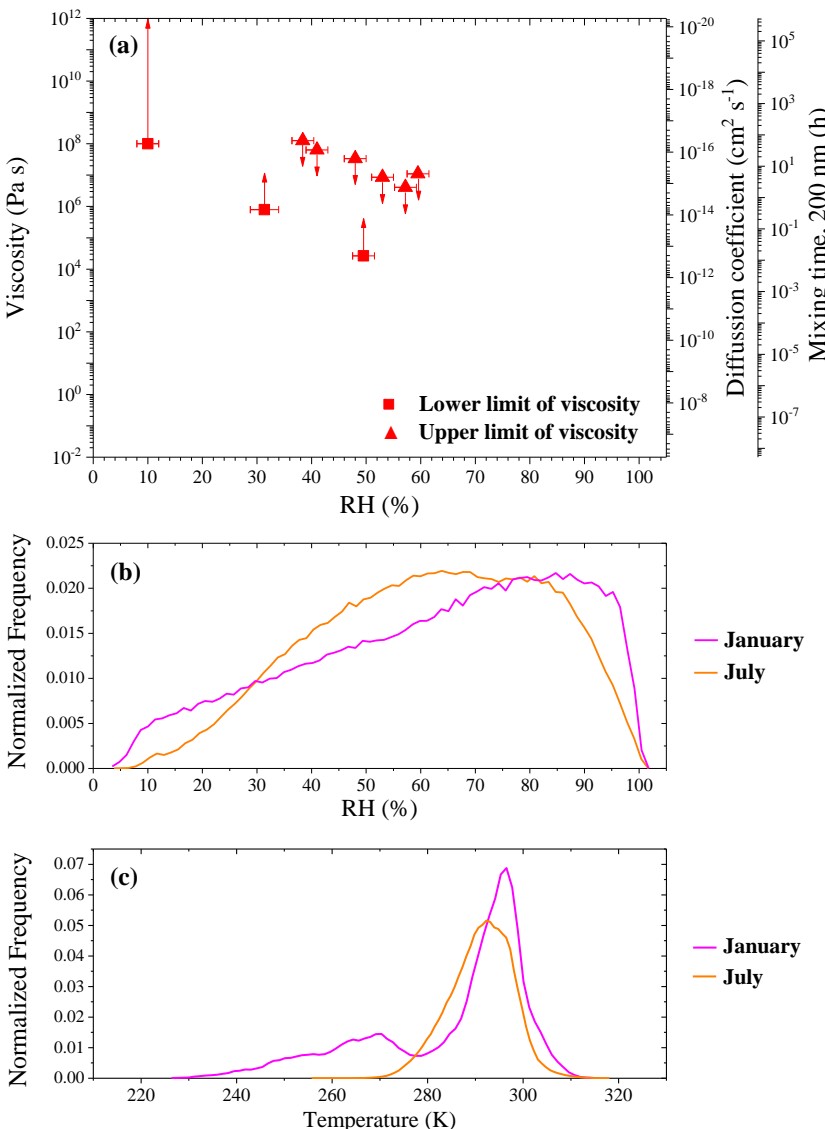

Figure 5. Panel (a): Viscosities, diffusion coefficients, and mixing times of organic molecules
within 200nm diesel fuel SOA. Panel b and c represent the RH frequency distribution and the
temperature frequency distribution in the planetary boundary layer when the average
concentrations of organic aerosol are higher than 0.5 μg m$^{-3}$ at the surface based on GEOS-
Chem (Ullmann et al., 2018). The frequency distributions were calculated using monthly mean





1    meteorological data from GEOS-Chem version v10-01 and data was only included when the

2    monthly mean concentrations of organic aerosol at the surface were greater than 0.5 μg m$^{-3}$

3    (Maclean et al., 2017).