# Peer review of "Liquid-liquid phase separation and viscosity within secondary organic"

_Atmospheric Chemistry and Physics, 2019_

## Referee Comment (RC1) · Anonymous Referee #1 · 28 May 2019

Th manuscript describes the analysis of secondary organic aerosol (SOA) samples generated from diesel fuel vapours, concentrating on observations of liquid-liquid phase separation and viscosity. This continues a sequence of publications on similar topics by the authors examining a variety of SOA systems, expanding the coverage of atmospheric aerosol surrogate systems that can improve our understanding of ambient particles.

The authors should consider my specific comments below in revising their manuscript.

Collection of samples on glass slide for liquid-liquid phase separation and viscosity measurements: It would be helpful if the authors could say something about the possible changes in composition (particularly partitioning of semi-volatile and more volatile components) that result – how reflective of the actual aerosol composition in the Teflon chamber are the particles once deposited and then analysed on the surface? This may be discussed in previous work but should be reiterated here. For example, the viscosity measurements are made in a flow-cell and necessarily this will presumably drive the composition to lower volatility and higher viscosity components over time. These changes in composition are not accounted for in the predictions of viscosity based on mass spectrometry measurements of the SOA composition.

Page 8: The authors state "The increase in the range of RH values over which LLPS occurs is likely related to the spread in O:C values within the organic particles – as the spread in O:C values increases, the RH range for LLPS is also likely to increase." In itself, the spread is not a reason, is it? Can the authors provide some rationale for why increasing chemical complexity leads to an increase in the LLPS RH range?

Figure 1: Are (a-c) just three different particles prepared under the same reaction chamber conditions? Similarly, in Figure 2, the different poke-flow measurements are made on different particles?

Page 8 onwards: Lower and upper limits to viscosity could clearly be dependent on any changes in composition that occur during the conditioning period. It would be helpful if the authors could show how the measured viscosity changes during the conditioning period, does it tend to a limit as the conditioning time increases? Is this conditioning based solely on adjustment to RH or is there some change in composition/partitioning of components into the gas phase during this time period (given the high concentrations in the reactor chamber)?

Page 9: To what extent is it appropriate to even represent the viscosity by a single value given that could be multiple phases/heterogeneities with within the particle with different viscosities? For example, the method for recording the "upper limit" could presumably be a measure of the viscosity of one phase, providing sufficient mobility for

the shape recovery, with a more viscous phase moving within the more mobile phase.

Page 12: The authors state "At 30 % RH $\tau$mixing is âĽş0.4 h, and at 38 to 50 % RH $\tau$mixing is in the range of $\sim$0.01 h to $\sim$50 h (Fig. 5a). These results provide important constraints on $\tau$mixing values within anthropogenic SOA." Given the viscosity goes down with increasing RH, this sentence is confusing. The scales on Figure 5(a) are confusing, partly because it is very hard to read values due to the extremely large range. I recommend the authors only show a viscosity range from 104 Pa s to $\sim$109 Pa s. Lower and higher viscosities are to a large extend superfluous and there are no data in these arranges.

---

## Referee Comment (RC2) · Anonymous Referee #2 · 14 Jun 2019

In this study the authors investigate liquid-liquid phase separation (LLPS) and viscosity of secondary organic aerosol (SOA) formed in photooxidation of diesel fuel vapors. Such multi-precursors studies on SOA are needed to understand atmospheric SOA and the topic fits well in the scope of Atmospheric Chemistry and Physics. The study is based on experimental observations of LLPS and viscosity and comparison to viscosity parameterizations. Overall the study seem carefully conducted. The manuscript is in most parts clear and carefully written, although I have listed below few points which should be considered before the manuscript is accepted for publication.

Major comment:

[Figure]

While the authors have overall resented the results carefully considering uncertainties and they discuss some caveats of the technique on page 12 L4-10, one aspect of the experimental technique has been neglected in the discussion of the results. The particles where collected on the glass slides and then conditioned for up to an hour for desired RH. During the conditioning (and the viscosity/LLPS measurements), the particles were surrounded by gas phase free of organics and this should cause some of the organics to evaporate. Therefore, the composition of the particles during viscosity or LLPS measurements would not have been the same as for it was when the particles were suspended in the chamber. This issue should be mentioned and the resulting uncertainty or systematic error in viscosities should be discussed in the text.

Minor comments:

P7 L1-8: Why are different values of coefficient k_GT used?

P7 L13-20: Is the predicted viscosity sensitive for the assumed viscosity of $10^{-5}$ at very high temperature and the assumed value of $D_f = 10$? Why did the authors chose $D_f = 10$ which they state is a lower limit from a previous study? Was the viscosity parameterization tuned to match the measured values in this study by selecting these specific values?

P8 L6-7: "two liquid phases may still have been present in the particles, but not in the focus of the microscope" Meaning of this sentence was not clear, probably since the authors have not explained how these optical measurements where actually done. What is the difference between the particles a-c in Fig. 1 and particles a-c in Fig. S2? Where all of these particles generated under same conditions? It seems that there is LLPS at 70 % RH in particles in Fig. 1, but in particles shown in Fig. S2 there is no LLPS always even at ∼82% RH. What causes this? Fig. S2 c shows LLPS at 4.7 % RH but not at the higher RH of 50.2%. This raises a question about the reliability of these results and such issues should be explained in the manuscript.

Technical comments:

P12 L18: "the presence of an organic-rich outer phase at high RH values can lower the barrier to cloud droplet formation" Barrier of what? Please rephrase.

Fig. 3b: Please revise the legend regarding Song et al. results. Currently the legend advises that Song et al. results would be shown with a green solid line. As such line is not present in the figure, I assume these results are the ones shown by the green errorbars.

Fig. 4: The times written on top of the figures do not show well. Please consider writing them with another color.

Fig. S1: "Mass Dp" and "particle Dp" where confusing in labels. The "Mass Dp" refers also to the diameter of particles. E.g. "Mass Dp" and "Number Dp" would be more clear and consistent names.

---

## Author Comment (AC1) · 25 Jul 2019

**Referee #1**

Summary:
The manuscript describes the analysis of secondary organic aerosol (SOA) samples generated from diesel fuel vapours, concentrating on observations of liquid-liquid phase separation and viscosity. This continues a sequence of publications on similar topics by the authors examining a variety of SOA systems, expanding the coverage of atmospheric aerosol surrogate systems that can improve our understanding of ambient particles. The authors should consider my specific comments below in revising their manuscript.

Concerns:
*[1]* Collection of samples on glass slide for liquid-liquid phase separation and viscosity measurements: It would be helpful if the authors could say something about the possible changes in composition (particularly partitioning of semi-volatile and more volatile components) that result – how reflective of the actual aerosol composition in the Teflon chamber are the particles once deposited and then analyzed on the surface? This may be discussed in previous work but should be reiterated here. For example, the viscosity measurements are made in a flow-cell and necessarily this will presumably drive the composition to lower volatility and higher viscosity components over time. These changes in composition are not accounted for in the predictions of viscosity based on mass spectrometry measurements of the SOA composition.

*[A1]* This is a good question, and we have addressed it in previous publications, but we agree that we should also address this question in the current manuscript. To address the referee's comments we will add the following to the revised manuscript (Sect. 2.3).
"In the poke-and-flow experiments (as well as the LLPS experiments), the particles are exposed to a constant flow of gas which can lead to a change in the composition of the particles by partitioning of semi-volatiles to the gas phase. For a 1 hr poke-and-flow experiment, the amount of gas exposed to the SOA is 30 L compared to 380 L collected from the environmental chamber. Exposing the SOA to this amount of gas can be considered equivalent to changing the mass loading used to generate the SOA from 550 μg m$^{-3}$ to 510 μg m$^{-3}$. Exposing the particles to a constant gas flow for 27 hours (maximum amount of time a sample was exposed to a constant gas flow) can be considered equivalent to changing the mass loading from 550 μg m$^{-3}$ to 175 μg m$^{-3}$. This should be considered a worse-case scenario since this estimation does not consider kinetic constraints to evaporation. Based on previous measurements, the viscosity of toluene SOA is independent of mass loadings ranging from ~800 μg m$^{-3}$ to ~80 μg m$^{-3}$ (Song et al., 2016a). Assuming that diesel fuel SOA behaves like toluene SOA, the viscosity of diesel fuel SOA should not be influenced by exposure to a constant flow of gas in our poke-and-flow experiments. Consistent with this discussion, we did not observe a relationship between particle viscosity and time the SOA was exposed to a constant flow of gas in our experiments."

Reference:
Song, M., Liu, P. F. F., Hanna, S. J., Zaveri, R. A., Potter, K., You, Y., Martin, S. T., and Bertram,

A. K.: RH-dependent viscosity of secondary organic material from toluene photo-oxidation and possible implications for organic particulate matter over megacities, Atmos. Chem. Phys., 16, 8817-8830, 10.5194/acp-16-8817-2016, 2016a.

*[2]* Page 8: The authors state "The increase in the range of RH values over which LLPS occurs is likely related to the spread in O:C values within the organic particles – as the spread in O:C values increases, the RH range for LLPS is also likely to increase." In itself, the spread is not a reason, is it? Can the authors provide some rationale for why increasing chemical complexity leads to an increase in the LLPS RH range?

*[A2]* We will re-write this section for clarity (Sect. 3.1). Specially the following will be added to the manuscript in place of the discussion on spread in O:C. "The increase in the range of RH values over which LLPS occurs is likely related to distribution of the polarities (or hydrophilicities) of the organics molecules within the SOA (Renbaum-Wolff et al., 2016; Gorkowski et al., 2019). When the organic molecules are hydrophobic or moderately hydrophobic (and hence have small O:C values) the particles are expected to have a single organic-rich phase until close to 100% RH, at which point LLPS can occur. When the organic molecules are hydrophilic (and hence have large O:C values), the particles are expected to have a single water-rich phase, with no occurrence of LLPS. Alternatively, if the particles contain a mixture of hydrophobic and hydrophilic organic molecules, the particles are expected to have both an organic-rich phase and a water-rich phase over a relatively wide range of RH values. A significant amount of molecules with low and high O:C values in the diesel SOA studied here (Fig. S3) is consistent with LLPS being observed over a relatively wide range of RH values."

[Figure]

Figure S3. Distribution of O:C values and carbon numbers of the organic molecules in the diesel fuel SOA studied here. The size of the symbols indicates the relative amount of the organic molecules in the SOA based on the ion current in the mass spectrum.

References:

Gorkowski, K., Preston, T. C., and Zuend, A.: RH-dependent organic aerosol thermodynamics via an efficient reduced-complexity model, Atmos. Chem. Phys. Discuss., https://doi.org/10.5194/acp-2019-495, in review, 2019.

Renbaum-Wolff, L., Song, M., Marcolli, C., Zhang, Y., Liu, P. F., Grayson, J. W., Geiger, F. M., Martin, S. T., and Bertram, A. K.: Observations and implications of liquid-liquid phase separation at high relative humidities in secondary organic material produced by α-pinene ozonolysis without inorganic salts, Atmos. Chem. Phys., 16, 7969–7979, https://doi.org/10.5194/acp16-7969-2016, 2016.

*[3]* Figure 1: Are (a-c) just three different particles prepared under the same reaction chamber conditions? Similarly, in Figure 2, the different poke-flow measurements are made on different particles?

*[A3]* Yes, Fig. 1 and Fig. S2 show six different particles prepared under the same reaction chamber conditions. Also, a different particle, prepared with the same reaction conditions, was used for each poke-and-flow measurement. To address the referee's comments, this information will be added to the revised manuscript (Sect. 3.1) and the caption for Fig. 3.

*[4]* Page 8 onwards: Lower and upper limits to viscosity could clearly be dependent on any changes in composition that occur during the conditioning period. It would be helpful if the authors could show how the measured viscosity changes during the conditioning period, does it tend to a limit as the conditioning time increases? Is this conditioning based solely on adjustment to RH or is there some change in composition/partitioning of components into the gas phase during this time period (given the high concentrations in the reactor chamber)?

*[A4]* See response to [A1] above.

*[5]* Page 9: To what extent is it appropriate to even represent the viscosity by a single value given that could be multiple phases/heterogeneities with within the particle with different viscosities? For example, the method for recording the "upper limit" could presumably be a measure of the viscosity of one phase, providing sufficient mobility for the shape recovery, with a more viscous phase moving within the more mobile phase.

*[A5]* This is a good point. When calculating the viscosity, we did not take into account the heterogeneity of the particle (i.e. the presence of both an organic-rich and water-rich phase). The viscosity measurements were carried out at RH values $\lesssim 58\ \%$ RH. For this RH range, the amount of the water-rich phase was small but still detectable in most cases. Assuming the

water-rich phase is less viscous than the organic-rich phase, due to the plasticizing effect of water, the viscosity of the organic-rich phase will be greater than the calculated (i.e. reported) viscosities. To address the referee's comments, we will add this caveat to the revised manuscript (Sect. 3.3).

*[6]* Page 12: The authors state "At 30 % RH $\tau_{mixing}$ is $\gtrsim$0.4 h, and at 38 to 50 % RH $\tau_{mixing}$ is in the range of ~0.01 h to ~50 h (Fig. 5a). These results provide important constraints on $\tau_{mixing}$ values within anthropogenic SOA." Given the viscosity goes down with increasing RH, this sentence is confusing. The scales on Figure 5(a) are confusing, partly because it is very hard to read values due to the extremely large range. I recommend the authors only show a viscosity range from $10^4$ Pa s to ~$10^9$ Pa s. Lower and higher viscosities are to a large extend superfluous and there are no data in these arranges.

*[A6]* To address the referee's comments, the statement mentioned above will be changed to the following: "At 38 − 50 % RH $\tau_{mixing}$ are in the range ~0.01 h to ~50 h (Fig. 5a). These results provide important constraints on $\tau_{mixing}$ values within anthropogenic SOA." In addition, we will decrease the range of viscosities shown in Fig. 5a, for clarity.

---

## Author Comment (AC2) · 25 Jul 2019

**Referee #2**

Summary:
In this study the authors investigate liquid-liquid phase separation (LLPS) and viscosity of secondary organic aerosol (SOA) formed in photooxidation of diesel fuel vapors. Such multi-precursors studies on SOA are needed to understand atmospheric SOA and the topic fits well in the scope of Atmospheric Chemistry and Physics. The study is based on experimental observations of LLPS and viscosity and comparison to viscosity parameterizations. Overall the study seem carefully conducted. The manuscript is in most parts clear and carefully written, although I have listed below few points which should be considered before the manuscript is accepted for publication.

Major comment:

*[1]* While the authors have overall resented the results carefully considering uncertainties and they discuss some caveats of the technique on page 12 L4-10, one aspect of the experimental technique has been neglected in the discussion of the results. The particles where collected on the glass slides and then conditioned for up to an hour for desired RH. During the conditioning (and the viscosity/LLPS measurements), the particles were surrounded by gas phase free of organics and this should cause some of the organics to evaporate. Therefore, the composition of the particles during viscosity or LLPS measurements would not have been the same as for it was when the particles were suspended in the chamber. This issue should be mentioned and the resulting uncertainty or systematic error in viscosities should be discussed in the text.

*[A1]*   This is a similar comment as Referee #1. Please see the answer *[A1]*.

Minor comments:
*[2]* P7 L1-8: Why are different values of coefficient k_GT used?

*[A2]* Based on previous studies, $k_{GT}$ of 1 is applied for organic-organic mixtures and $k_{GT}$ of 2.5 is applied for organic-water mixtures. We will clarify this point in the revised manuscript in Sect. 2.4.

*[3]* P7 L13-20: Is the predicted viscosity sensitive for the assumed viscosity of $10^{-5}$ at very high temperature and the assumed value of $D_f = 10$? Why did the authors chose $D_f = 10$ which they state is a lower limit from a previous study? Was the viscosity parameterization tuned to match the measured values in this study by selecting these specific values?

*[A3]* The viscosity of $10-5$ Pa s at a very high temperature is well established in the glass community (Angell, 1991; Angell, 2002). In these equations, Df is the fragility parameter and T0 is the Vogel temperature. In our calculations, we fixed Df to be 10 because a previous study that showed Df approaches 10 when the molar mass of the organic compounds exceed ~200 g

mol-1 (DeRieux et al., 2018) and because many of the detected compounds in diesel SOA have molar masses > 200 g mol-1. Even though the Df value does affect predicted viscosity (see Fig. 5b in DeRieux et al., 2018), Df is not as critical as other parameters such as the glass transition temperature or hygroscopicity. We will clarify this point in the revised manuscript in Sect. 2.4.

References:

Angell, C. A.: Relaxation in liquids, Polymers and plastic crystals - Strong fragile patterns and problems, J. Non-Cryst. Solids, 131, 13-31, https://doi.10.1016/0022-3093(91)90266-9, 1991.

Angell, C. A.: Liquid fragility and the glass transition in water and aqueous solutions, Chem. Rev., 102, 2627-2649, UNSP CR000689Q 10.1021/cr000689q, 2002.

DeRieux, W. S., Li, Y., Lin, P., Laskin, J., Laskin, A., Bertram, A. K., Nizkorodov, S. A., and Shiraiwa, M.: Predicting the glass transition temperature and viscosity of secondary organic material using molecular composition, Atmos. Chem. Phys., 18, 6331-6351, 10.5194/acp-18-6331-2018, 2018.

*[4]* P8 L6-7: "two liquid phases may still have been present in the particles, but not in the focus of the microscope" Meaning of this sentence was not clear, probably since the authors have not explained how these optical measurements where actually done.

*[A4]* To address this question, the following information on how the optical measurements were done will be included in the revised manuscript in Sect. 2.2:

"SOA was collected on hydrophobic glass slides by impaction, resulting in SOA particles on the hydrophobic glass slides with diameters > 10 µm and a spherical cap geometry. LLPS was detected using an optical microscope (Zeiss Epiplan 10X/0.20 HD) coupled to a flow-cell with temperature and RH control (Parsons et al., 2004; Pant et al., 2006; Song et al., 2012b). During the experiments, a constant flow (1.5 L min$^{-1}$) of humidified $N_2$ gas was maintained within the flow-cell and measured with a dew point hygrometer (General Eastern M4/E4 Dew Point Monitor, Canada). The temperature within the flow-cell was maintained at 290 ± 1 K and measured with a thermocouple (OMEGA, Canada). At the beginning of the experiments, the SOA particles were equilibrated at around 100 % RH for at least 15 min. At this point, the focus of the microscope was adjusted so the focal plane of the microscope corresponded to the top or interior of several SOA particles. Due to the different sizes of the SOA particles on the hydrophobic glass slides, the focal plane of the microscope corresponded to the top of some SOA particles and the middle of some SOA particles while some smaller particles were not in the focal plane (leading to blurry images). Next, the RH was reduced at a rate of 0.5% RH min$^{-1}$ until a value close to 0% was reached. While the RH was decreased, images of the particles were acquired every 10 sec with a CCD camera connected to the microscope. From the images, the number of phases (e.g. one phase or two phases) present in the particles were determined. Typically the focus of the microscope was not adjusted as the RH was reduced. As the RH was reduced, the size of the SOA particles decreased due to the loss of water, and some SOA particles that were in focus at high RH values became out of focus at low RH values."

*[5]* What is the difference between the particles a-c in Fig. 1 and particles a-c in Fig. S2? Where all of these particles generated under same conditions?

*[A5]* Figs. 1 and S2 show six different particles generated with the same conditions. This information will be added to the revised manuscript (Sect. 3.1).

*[6]* It seems that there is LLPS at 70 % RH in particles in Fig. 1, but in particles shown in Fig. S2 there is no LLPS always even at 82% RH. What causes this? Fig. S2 c shows LLPS at 4.7 % RH but not at the higher RH of 50.2%. This raises a question about the reliability of these results and such issues should be explained in the manuscript.

*[A6]* This confusion is mainly from us not illustrating well the morphology of the particles in Figs. 1 and S2. In the original manuscript, the illustrations suggested that there is no LLPS in Fig. 1c, 5.7%; Fig. S2b, 82.6%; Fig. S2b, 74.1%; Fig S2b, 52.0%; and Fig. S2c, 50.2%. After closer inspection, the images corresponding to these illustrations do have heterogeneity (i.e. non-uniformity) that suggests the presence of a small amount of a second phase. In the revised manuscript, we will adjust the illustrations in Figs. 1 and S2 and corresponding text to make this point clear.
In addition, the poor quality of the images in Fig. S2a make it impossible to determine if there is LLPS in this particle at $\leq 70.9\%$; the images are extremely blurry and the particle is outside the focal plane at RH values $\leq 70.9\%$. As a result, we will remove Fig. S2a from the revised manuscript. Sorry for the confusion we created by include these poor quality images.

Technical comments:

*[7]* P12 L18: "the presence of an organic-rich outer phase at high RH values can lower the barrier to cloud droplet formation" Barrier of what? Please rephrase.
*[A7]* This sentence will be changed to the following: "… since the presence of an organic-rich outer phase at high RH values can lower the supersaturation with respect to water required for cloud droplet formation."

*[8]* Fig. 3b: Please revise the legend regarding Song et al. results. Currently the legend advises that Song et al. results would be shown with a green solid line. As such line is not present in the figure, I assume these results are the ones shown by the green error bars.
*[A8]* The line corresponding to Song et al. will be changed to a circle and a circle will be added to the data for clarity.

*[9]* Fig. 4: The times written on top of the figures do not show well. Please consider writing them with another color.
*[A9]* As suggested, we will change the color to white in Fig. 4.

*[10]* Fig. S1: "Mass Dp" and "particle Dp" where confusing in labels. The "Mass Dp" refers also to the diameter of particles. E.g. "Mass Dp" and "Number Dp" would be more clear and consistent names.

*[A10]* Thank you for the suggestion. We will change the names to Mass Dp and Number Dp as suggested.

---

## Author Response (AR2)

Professor Annele Virtanen,
Co-Editor of Atmospheric Chemistry and Physics
 annele.virtanen@uef.fi

Dear Annele,

Below are our responses to the comment provided by reviewer #2. For clarity, the referee comments or questions are in black text, and are preceded by bracketed, italicized numbers (e.g. *[1]*). Our responses are in in blue text below each comment or question with matching italicized numbers (e.g. *[A1]*). We thank the referee for their time and care reading our manuscript and for their helpful comments and questions.

Sincerely,

Allan Bertram
Professor of Chemistry
University of British Columbia

**Referee #2**
Comments:

*[1]* To address my comment and the first comment by the referee 1 the authors have modified the manuscript by adding discussion on evaporation of compounds during conditioning. I have concerns regarding the calculations the authors present for this issue and the conclusions they draw from them. I recommend a revision before the manuscript is accepted.

Authors state: "In the poke-and-flow experiments (as well as the LLPS experiments), the particles are exposed to a constant flow of gas which can lead to a change in the composition of the particles by partitioning of semi-volatiles to the gas phase. For a 1 hr poke-and-flow experiment, the amount of gas exposed to the SOA is 30 L compared to 380 L collected from the environmental chamber. Exposing the SOA to this amount of gas can be considered equivalent to changing the mass loading used to generate the SOA from 550 $\mu$g m$^{-3}$ to 510 $\mu$g m$^{-3}$. Exposing the particles to a constant gas flow for 27 hours (maximum amount of time a sample was exposed to a constant gas flow) can be considered equivalent to changing the mass loading from 550 $\mu$g m$^{-3}$ to 175 $\mu$g m$^{-3}$. This should be considered a worse-case scenario since this estimation does not consider kinetic constraints to evaporation. Based on previous measurements, the viscosity of toluene SOA is independent of mass loadings ranging from ~800 $\mu$g m$^{-3}$ to ~80 $\mu$g m$^{-3}$ (Song et al., 2016a). Assuming that diesel fuel SOA behaves like toluene SOA, the viscosity of diesel fuel SOA should not be influenced by exposure to a constant flow of gas in our poke-and-flow experiments. Consistent with this discussion, we did

not observe a relationship between particle viscosity and time the SOA was exposed to a constant flow of gas in our experiments."

This does not seem valid reasoning. This would approximately hold if adding 30 L of clean air to a 380 L expansible chamber where mass loading where initially 550 μg m$^{-3}$. However, according to my understanding, in this study the authors collect the particles on a glass slide and then clean air is flowing above this glass slide during conditioning. In this case the equilibration calculation presented above does not hold and the system to consider is the particle phase formed by the particles on the glass slide and the organic-free clean gas flowing above them. As the particles are surrounded by the clean air continuously, they are evaporating. Given enough time they should actually evaporate completely. Therefore the presented calculation is not a worse-case scenario like the authors stated. The observed lack of a relationship between viscosity and the time the particles were exposed to the gas flow could be due to the relatively long exposure time even in the fastest case: 1 hour is enough time for semi-volatiles to evaporate considerably. Therefore the particle composition could be changing during the conditioning in all of the experiments to the extent that it does not represent the viscosity of the original particles. The issue of evaporating compounds and the resulting uncertainty or systematic error in viscosities should be discussed in the text.

*[A1]* Thank you for feedback. We do think our equilibrium calculations are useful to put in context the amount of "dilution" that occurs in our experiments, but we concede that the equilibrium calculations do not exactly represent our experiments. As suggested by the referee, we removed the equilibrium calculations from the manuscript and modified the manuscript to include the issue of evaporating compounds and the resulting uncertainty or systematic error in viscosities.

The following is the modified text in Section 2.3:

"We acknowledge that viscosity of the SOA could change between the time of its initial formation in the chamber and the time of the off-line viscosity and LLPS measurements. Such changes can be driven by both evaporative losses and slow chemical aging processes. During the MOUDI sampling, impacted particles are surrounded by the same gaseous products as in the chamber and should not evaporate. We expect the evaporation to be minimal when the collected particles are briefly exposed to ambient air, sealed in a storage container with a small head space volume (~ 2 cm$^3$), and frozen for storage and shipment. However, in the poke-and-flow experiments (as well as the LLPS experiments), the particles are exposed to a constant flow of purified air at room temperature for an extended period of time which can lead to a change in the composition of the particles by partitioning of semi-volatiles to the gas phase. For a 1 hr poke-and-flow experiment, the amount of gas exposed to the SOA is 30 L compared to 380 L collected from the environmental chamber. For a 27 hr poke-and-flow experiment (maximum amount of time a sample was exposed to a constant gas flow), the amount of gas exposed to the SOA was 810 L compared to 380 L collected from the environmental chamber. We did not observe a relationship between particle viscosity and time the SOA was exposed to

a constant flow of gas in our experiments; however, semi-volatiles may still have evaporated in the experiments when the particles were conditioned to a known RH and during the poke-and-flow measurements. The loss of semi-volatiles would lead to an increase in viscosity of the SOA (Wilson et al., 2015; Yli-Juuti et al., 2017; Buchholz et al., 2019). Consequently, our results should be considered as upper limits to the viscosity of the SOA generated with a particle mass loading of 550 µg m$^{-3}$. The evaporation of semi-volatiles in the experiments as well as possible slow chemical aging reactions occurring during shipment and storage may in fact have resulted in the SOA being more similar to the chemical composition of SOA in the atmosphere, which are formed at particle mass loadings < 550 µg m$^{-3}$ and then chemically aged."

The following is the text added to Section 3.3:
"Second, some of the semi-volatiles may have evaporated from the diesel fuel SOA in the poke-and-flow experiments. If evaporation of semi-volatiles did occur, the viscosity of the diesel fuel SOA at (~500 µg m$^{-3}$) may be lower than reported here (Wilson et al., 2015; Yli-Juuti et al., 2017; Buchholz et al., 2019)."

References:

Buchholz, A., Lambe, A. T., Ylisirnio, A., Li, Z. J., Tikkanen, O. P., Faiola, C., Kari, E., Hao, L. Q., Luoma, O., Huang, W., Mohr, C., Worsnop, D. R., Nizkorodov, S. A., Yli-Juuti, T., Schobesberger, S., and Virtanen, A.: Insights into the O : C-dependent mechanisms controlling the evaporation of alpha-pinene secondary organic aerosol particles, Atmos. Chem. Phys., 19, 4061-4073, 10.5194/acp-19-4061-2019, 2019.

Wilson, J., Imre, D., Beranek, J., Shrivastava, M., and Zelenyuk, A.: Evaporation Kinetics of Laboratory-Generated Secondary Organic Aerosols at Elevated Relative Humidity, Environ. Sci. Technol., 49, 243-249, 10.1021/es505331d, 2015.

Yli-Juuti, T., Pajunoja, A., Tikkanen, O. P., Buchholz, A., Faiola, C., Vaisanen, O., Hao, L. Q., Kari, E., Perakyla, O., Garmash, O., Shiraiwa, M., Ehn, M., Lehtinen, K., and Virtanen, A.: Factors controlling the evaporation of secondary organic aerosol from alpha-pinene ozonolysis, Geophys. Res. Lett., 44, 2562-2570, 10.1002/2016gl072364, 2017.